

# Linear stability of Einstein and de Sitter universes in the quadratic theory of modified gravity

**Mudhahir Al Ajmi**⋆

Department of Physics, College of Science, Sultan Qaboos University,
P.O. Box 36, P.C. 123, Muscat, Sultanate of Oman

⋆ mudhahir@squ.edu.om

## Abstract

**We consider the Einstein static and the de Sitter universe solutions and examine their instabilities in a subclass of quadratic modified theories for gravity. This modification proposed by Nash is an attempt to generalize general relativity. Interestingly, we discover that the Einstein static universe is unstable in the context of the modified gravity. In contrast to Einstein static universe, the de Sitter universe remains stable under metric perturbation up to the second order.**

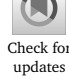

## 1 Introduction

According to Wilkinson Microwave anisotropy probe [1, 2], the BICEP2 experiment [3, 4], Sloan Digital Sky Surveys [5] and Planck satellite [6–8], it turns out that less than 5% of the Universe is composed of ordinary matter, 68% is of dark energy and 27% of Universe is composed of the dark matter. On the other hand it has been shown also that the observed Universe is undergoing an accelerated expansion [9–13]. The late-time cosmic acceleration can be explained by two promising explanations, at least. One of them is the dark energy component in the Universe [14] deduced from the abovementioned detectors although the nature of the dark energy is not known yet. The other explanation is to tackle the problem using a geometrical picture modifying Einstein theory of gravity. The approach is known as the modified gravity which have several motivations in high-energy physics, cosmology and astrophysics [15, 16]. Modified theories of gravity can be achieved from different contexts. In modifying gravity theories $f(R), f(R, T), f(G), f(R, G), f(R, \varphi)$ and $f(R, R_{ab}R^{ab}, \varphi)$ are some attractive choices. Here, $R$, $T$, $G$, $R_{ab}R^{ab}$ and $\varphi$ are Ricci scalar, trace of energy momentum tensor, Gauss-Bonnet invariant, Ricci invariant and scalar field respectively. In $f(R)$ theories of gravity the Lagrangian density $f$ is an arbitrary function of the scalar curvature $R$ [17, 18].

One of the earlier modifications to Einstein's general relativity (GR) is known as the Brans-Dicke gravity. This theory introduced a dynamical scalar field using a variable gravitational constant [19]. Later, there was a study of a scalar-tensor theory of gravity in which the metric is coupled to a scalar field where, a 'missing-mass problem' can be successfully described [20]. This approach can be applied to the Bianchi cosmological models.

These theories, also, include higher order curvature invariants [21,22]. For example, some modified gravity theory models describe flat rotation of galaxies without taking into account the cold dark matter particles [24]. Other models which describe accelerating Universe expansion with a quadratic term of $R$ in the Lagrangian density, proposed by Starobinsky [23].

In these theories it is an important task to show and prove that these Universe models are stable against small perturbations in the Hubble parameters. Searches have been performed showing that Einstein Universe [25], for example, is stable against such perturbations in vector or tensor field or scalar density instabilities.

John Nash has developed Einstein's theory of gravity alternatively. Quantum gravity theories can make a benefit using this theory. Also, many cosmological models have a problem of divergence whereas this theory is divergence free.

In this paper, we examine the Einstein static and the de Sitter Universe solutions and quantify their stabilities for the Bianchi type I model using Nash modified theory of gravity in the context of Bianchi Universe geometry.

In Section II we introduce the Nash theory for gravity. In Section III we explain the Bianchi-I Universe and in Section IV the Lagrangian used in the paper is stated. In Section V we apply the Lagrangian with respect to Einstein static Universe and study its corresponding stability. In Section IV we repeat the analysis in Section V but with the de Sitter Universe. Finally, we conclude our findings in the last section.

## 2   Nash's theory for gravity

J. Nash developed an alternative theory of modified gravity in which he modified the GR. This is a way to ultimately consider GR as renormalizable theory [26]. More recent works are presented in [27–29]. The original Nash gravity action proposed without Einstein-Hilbert general relativity term is:

$$\mathcal{S} = \int d^4x \mathcal{L} = \int d^4x \sqrt{-g}\Big(2R^{\mu\nu}R_{\mu\nu} - R^2\Big). \tag{1}$$

There are several Lagrangians used to develop theories of quantum gravity. The one written above is one of them. Using the above action and taking into account the metric $g^{\mu\nu}$ as a dynamical field, the gravitational field equations are directly derived as:

$$\Box G_{\mu\nu} + G_{\alpha\beta}\Big(2R^{\alpha\beta}_{\mu\nu} - \frac{1}{2}g_{\mu\nu}R^{\alpha\beta}\Big) = 0. \tag{2}$$

Nash theory have been investigated in the context of Noether thery and its cosmological implication [30]. In this project we examine the cosmological solutions for homogeneous anisotropic Universe.

## 3   Bianchi-I Universe

Due to the diversity and non-locality of subregions in the Universe in terms of galactic distributions and internal galactic structures we study Nash model in the context of Bianchi Type

I model which can considers the homogeneity and anisotropy as well as the non-rotation of cosmos. We recall that in FLRW model the scale factor is unique. However, because of the anisotropic feature of the Bianchi Type I different scale factors are encountered in each direction. Consequently, in $x^\mu = (t, x, y, z)$ coordinates, the line-element is,

$$ds^2 = -dt^2 + A^2(t)dx^2 + B^2(t)dy^2 + C^2(t)dz^2. \tag{3}$$

Here, $A(t)$, $B(t)$ and $C(t)$ are scale factors and they are all functions of the cosmic time, $t$. It is notable that the FLRW has been generalized to the above metric function. The mean of the three directional Hubble parameters in the Bianchi Type I Universe is given by $H = \frac{1}{3}\sum H_i$ where $H_i = \frac{d\ln(A_i)}{dt}$, $A_i = \{A, B, C\}$.

## 4 Point-like Lagrangian

Here, the Lagrangian can be formalized as point-like parameters characterized by the configuration space, i.e. $\mathcal{L} = \mathcal{L}(A, B, C, H_1, H_2, H_3, \dot{H}_1, \dot{H}_2, \dot{H}_3)$.

We define the Hubble constants as new parameters:

$$H_1 = \frac{\dot{A}}{A} = \frac{d\ln(A(t))}{dt}, \quad H_2 = \frac{\dot{B}}{B} = \frac{d\ln(B(t))}{dt}, \quad H_3 = \frac{\dot{C}}{C} = \frac{d\ln(C(t))}{dt}, \tag{4}$$

where the unknown functions which must be obtained by this symmetry methods are $\{H_1, H_2, H_3\}$.

We consider the metric (3) and substitute it into the action (1). We then perform the integration by parts and eliminate the second derivative terms with respect to time ($\ddot{A}_i$). We, then, obtain the following point-like Lagrangian, which enables us to investigate the symmetry properties of the system:

$$\begin{aligned}
\mathcal{L} = \frac{-4}{ABC}\Big( & H_1^3 H_2 + H_1^3 H_3 + H_1 H_2^3 + H_1 H_3^3 + H_2^3 H_3 + H_2 H_3^3 \\
& + H_1^2 H_2^2 + H_1^2 H_3^2 + 3 H_1^2 H_2 H_3 + 3 H_1 H_2^2 H_3 + 3 H_1 H_2 H_3^2 \\
& + H_1^2 \dot{H}_2 + H_1^2 \dot{H}_3 + H_2^2 \dot{H}_1 + H_2^2 \dot{H}_3 + H_3^2 \dot{H}_1 + H_3^2 \dot{H}_2 \\
& + H_1 H_2 \dot{H}_1 + H_1 H_2 \dot{H}_2 + H_1 H_3 \dot{H}_1 + H_1 H_3 \dot{H}_3 + H_2 H_3 \dot{H}_2 + H_2 H_3 \dot{H}_3 \\
& + 2 H_1 H_2 \dot{H}_3 + 2 H_1 H_3 \dot{H}_2 + 2 H_2 H_3 \dot{H}_1 + \dot{H}_2 \dot{H}_1 + \dot{H}_3 \dot{H}_1 + \dot{H}_3 \dot{H}_2 \Big).
\end{aligned} \tag{5}$$

## 5 Einstein static solutions and their (in)stability

The viability of cosmological solutions is plagued by the issue of stability. This is so because of the existence of varieties of perturbations, e.g. the quantum fluctuations. In order to study the instability problem, the Einstein static solutions must be retrieved in the context of different theories of modified gravity [31–43]. In this section, we examine the stability of the solutions by using a local stability method. Using the metric (3), the field equations for Nash gravity can be written in the following system of second order differential equations:

$$\begin{aligned}
(& H_2^3 + H_3^3 + 3 H_1^2 H_2 + 3 H_3 H_1^2 + 2 H_2^2 H_1 + 6 H_3 H_1 H_2 + 2 H_3^2 H_1 + 3 H_3 H_2^2 \\
& + \dot{H}_2 H_1 + \dot{H}_3 H_1 - \dot{H}_3 H_2 - 2 H_2 \dot{H}_3 - \dot{H}_3 H_3 - \ddot{H}_2 - \ddot{H}_3)ABC \\
& + (H_1 H_2 + H_1 H_3 + H_2^2 + 2 H_2 H_3 + H_3^2 + \dot{H}_2 + \dot{H}_3)(\dot{H}_3 AB'C + \dot{H}_3 ABC') \\
& - (\dot{H}_2 H_1^2 + \dot{H}_3 H_1^2 + \dot{H}_2 H_1 H_2 + 2 \dot{H}_3 H_1 H_2 + 2 \dot{H}_2 H_1 H_3 \\
& + \dot{H}_3 H_1 H_3 + \dot{H}_3 H_2^2 + \dot{H}_2 H_3 H_2 + \dot{H}_3 H_3 H_2 + H_3^2 \dot{H}_3 + \dot{H}_3 \dot{H}_2 + H)A'BC = 0,
\end{aligned} \tag{6}$$

and

$$
\begin{aligned}
(H_1^3 + H_3^3 &+ 2H_1^2 H_2 + 3H_3 H_1^2 + 3H_2^2 H_1 + 6H_3 H_1 H_2 + 3H_3^2 H_1 \\
&+ 3H_3 H_2^2 + 2H_3^2 H_2 - H_1 \dot{H}_1 + H_2 \dot{H}_1 + \dot{H}_3 H_2 - \dot{H}_3 H_3 - \ddot{H}_1 - \ddot{H}_3) ABC \\
&+ (H_1^2 + H_1 H_2 + 2H_1 H_3 + H_2 H_3 + H_3^2 + \dot{H}_1 + \dot{H}_3)(\dot{H}_1 A' BC + \dot{H}_3 ABC'), \\
&- (\dot{H}_3 H_1^2 + H_1 H_2 \dot{H}_1 + 2\dot{H}_3 H_1 H_2 + H_3 H_1 \dot{H}_1 + \dot{H}_3 H_1 H_3 + H_2^2 \dot{H}_1 + \dot{H}_3 H_2^2 \\
&+ 2H_3 H_2 \dot{H}_1 + \dot{H}_3 H_3 H_2 + H_3^2 \dot{H}_1 - \dot{H}_3 \dot{H}_1 + H) B' AC = 0,
\end{aligned}
\tag{7}
$$

as well as

$$
\begin{aligned}
(H_1^3 + H_2^2 &+ 3H_1^2 H_2 + 2H_3 H_1^2 + 3H_2^2 H_1 + 6H_3 H_1 H_2 + 3H_3^2 H_1 + H_2^3 \\
&+ 2H_3 H_2^2 + 3H_3^2 H_2 - H_1 \dot{H}_1 - \dot{H}_2 H_2 + H_3 \dot{H}_1 + \dot{H}_2 H_3 - \ddot{H}_1 - \ddot{H}_2) ABC \\
&+ (H_1^2 + 2H_1 H_2 + H_1 H_3 + H_2 H_3 + \dot{H}_1 + \dot{H}_2)(\dot{H}_1 A' BC + \dot{H}_2 AB'C) \\
&- (\dot{H}_2 H_1^2 + H_1 H_2 \dot{H}_1 + \dot{H}_2 H_1 H_2 + H_3 H_1 \dot{H}_1 + 2\dot{H}_2 H_1 H_3 + H_2^2 \dot{H}_1 \\
&+ 2H_3 H_2 \dot{H}_1 + \dot{H}_2 H_3 H_2 + H_3^2 \dot{H}_1 + H_3^2 \dot{H}_2 + \dot{H}_2 \dot{H}_1 + H) C' AB = 0,
\end{aligned}
\tag{8}
$$

where

$$
\begin{aligned}
H = H_1^3 H_2 &+ H_3 H_1^3 + H_2^2 H_1^2 + 3H_3 H_1^2 H_2 + H_3^2 H_1^2 \\
&+ H_2^3 H_1 + 3H_3 H_1 H_2^2 + 3H_3^2 H_1 H_2 + H_3^3 H_1 + H_2^3 H_3 + H_3^3 H_2^2 + H_3^3 H_2.
\end{aligned}
\tag{9}
$$

Here we have defined $A' = dA(t)/dH_1$, $B' = dB(t)/dH_2$ and $C' = dC(t)/dH_3$ and $A, B$ and $C$ are functions of $t$. The Einstein static universe is a closed universe and the scale factors $A(t), B(t), C(t)$ are constants, implying that,

$$
H_i = \dot{H}_i = 0,
\tag{10}
$$

where $i = 1, 2, 3$. Thus, we introduce perturbation in the Hubble parameter which depends only on time:

$$
H_i \rightarrow \delta H_i, \quad \dot{H}_i \rightarrow \delta \dot{H}_i, \quad \ddot{H}_i \rightarrow \delta \ddot{H}_i.
\tag{11}
$$

Substituting the perturbations (11) into Eqs.(6)-(8), we find that the perpetuated equations can be written as

$$
\delta \ddot{H}_2 + \delta \ddot{H}_3 = 0, \quad \delta \ddot{H}_1 + \delta \ddot{H}_3 = 0, \quad \delta \ddot{H}_1 + \delta \ddot{H}_2 = 0,
\tag{12}
$$

for which the solutions take the form

$$
\delta H_i = H_i^{(0)} \left( \frac{t}{t_0} \right) + H_i^{(1)},
\tag{13}
$$

where $H_i^{(0)}$ and $H_i^{(1)}$ are constants of integration. It is apparent from Eq.(12) that we have no oscillation equations. Consequently, the Einstein static universe is unstable against the perturbations and hence this concludes that Einstein universe is unstable in the context of the Nash gravity.

## 6  Stability of the de Sitter Universe

In the previous section we investigated the linear stability of the Einstein universe. The analysis was done quickly because in the Einstein Universe all Hubble terms and their time derivatives

vanish. In this section we will investigate the stability of the de Sitter solution. This Einstenian anisotropic metric is defined as the anisotropic expansion of the flat FLRW metric. The directional Hubble parameters $H_i = h_i$ are considered as stationary values at the beginning, when the space time metric is stationary. Plugging these constant Hubble parameters and $A = a e^{h_1 t}, B = b e^{h_2 t}, C = c e^{h_3 t}$ in the field equations (6-8) we obtain:

$$
\begin{aligned}
w A' B C + (h_2^3 + h_3^3 + 3 h_1^2 h_2 + 3 h_3 h_1^2 + 2 h_2^2 h_1 \\
+ 6 h_1 h_2 h_3 + 2 h_3^2 h_1 + 3 h_3 h_2^2) A B C = 0,
\end{aligned}
\tag{14}
$$

$$
\begin{aligned}
w B' A C + (h_1^3 + h_3^3 + 2 h_1^2 h_2 + 3 h_3 h_1^2 + 3 h_2^2 h_1 \\
+ 6 h_1 h_2 h_3 + 3 h_3^2 h_1 + 3 h_3 h_2^2 + 2 h_3^2 h_2) A B C = 0,
\end{aligned}
\tag{15}
$$

$$
\begin{aligned}
w C' A B + (h_1^3 + h_2^3 + 3 h_1^2 h_2 + 2 h_3 h_1^2 + 3 h_2^2 h_1 \\
+ 6 h_1 h_2 h_3 + 3 h_3^2 h_1 + 2 h_3 h_2^2 + 3 h_3^2 h_2) A B C = 0,
\end{aligned}
\tag{16}
$$

where

$$
\begin{aligned}
w \quad = \quad & -(h_1^3 h_2 + h_3 h_1^3 + h_2^2 h_1^2 + 3 h_3 h_1^2 h_2 + h_3^2 h_1^2 + h_2^3 h_1 + 3 h_3 h_1 h_2^2 + 3 h_3^2 h_1 h_2 \\
& + h_3^3 h_1 + h_2^3 h_3 + h_3^2 h_2^2 + h_3^3 h_2).
\end{aligned}
\tag{17}
$$

Now we make the perturbation $H_i \rightarrow h_i + \delta \xi_i(t)$ in equations (6-8). Using the constraint of the zero order equations (14-16) we obtain a system of the nonlinear differential equations for $\delta \xi_i(t)$ where we omit terms of high orders of $\delta \xi_i(t)$. Hence, we arrive at the following system of the differential equations:

$$
\begin{aligned}
& [(h_1 \delta \dot\xi_2) + (h_1 \delta \dot\xi_3) - (h_2 \delta \dot\xi_3) - (h_3 \delta \dot\xi_3) - 2(h_2 \delta \dot\xi_3) \\
& + (h_2^3 + 3 h_2^2 \delta \xi_2) + (h_3^3 + 3 h_3^2 \delta \xi_3) + 2(h_2^2 h_1 + 2 h_2 h_1 \delta \xi_2 + h_2^2 \delta \xi_1) \\
& + 2(h_3^2 h_1 + 2 h_3 h_1 \delta \xi_3 + h_3^2 \delta \xi_1) + 3(h_1^2 h_2 + 2 h_1 h_2 \delta \xi_1 + h_1^2 \delta \xi_2) \\
& + 3(h_3 h_1^2 + 2 h_3 h_1 \delta \xi_1 + h_1^2 \delta \xi_3) + 3(h_3 h_2^2 + 2 h_3 h_2 \delta \xi_2 + h_2^2 \delta \xi_3) \\
& + 6(h_3 h_1 h_2 + h_1 h_2 \delta \xi_3 + h_3 h_2 \delta \xi_1 + h_3 h_1 \delta \xi_2) - (\delta \ddot\xi_2) - (\delta \ddot\xi_3)] A B C \\
& + [h_2^2 + h_3^2 + h_1 h_2 + h_1 h_3 + 2 h_2 h_3](\delta \dot\xi_2 A B' C + \delta \dot\xi_3 A B C') \\
& - [h_1^2 \delta \dot\xi_2 + h_1^2 \delta \dot\xi_3 + h_2^2 \delta \dot\xi_3 + h_3^2 \delta \dot\xi_3 + \dot h_2 h_2 \delta \dot\xi_1 + h_1 h_2 \delta \dot\xi_2 + h_1 h_3 \delta \dot\xi_3 + h_3 h_2 \delta \dot\xi_2 + h_3 h_2 \delta \dot\xi_3 \\
& + 2(h_1 h_2 \delta \dot\xi_3 + h_1 h_3 \delta \dot\xi_2) - h] A' B C = 0,
\end{aligned}
\tag{18}
$$

and

$$
\begin{aligned}
& [-h_1 \delta \dot\xi_1 + h_2 \delta \dot\xi_1 - h_3 \delta \dot\xi_3 + h_2 \delta \dot\xi_3 + (h_1^3 + 3 h_1 \delta \xi_1) \\
& + 2(h_1^2 h_2 + 2 h_1 h_2 \delta \xi_1 + h_1^2 \delta \xi_2) + 2(h_3^2 h_2 + 2 h_3 h_2 \delta \xi_3 + h_3^2 \delta \xi_2) \\
& + 3(h_3 h_1^2 + 2 h_3 h_1 \delta \xi_1 + h_1^2 \delta \xi_3) + 3(h_2^2 h_1 + 2 h_2 h_1 \delta \xi_2 + h_2^2 \delta \xi_1) \\
& + 3(h_3^2 h_1 + 2 h_3 h_1 \delta \xi_3 + h_3^2 \delta \xi_1) + 3(h_3 h_2^2 + 2 h_3 h_2 \delta \xi_2 + h_2^2 \delta \xi_3) \\
& + 6(h_3 h_1 h_2 + h_1 h_2 \delta \xi_3 + h_3 h_2 \delta \xi_1 + h_3 h_1 \delta \xi_2) - \delta \ddot\xi_1 - \delta \ddot\xi_3] A B C \\
& + [h_1^2 + h_3^2 + h_1 h_2 + h_2 h_3 + 2 h_1 h_3](\delta \dot\xi_1 A' B C + \delta \dot\xi_3 A B C') \\
& - [h_1^2 \delta \dot\xi_3 + h_1 h_2 \delta \dot\xi_1 + 2 h_1 h_2 \delta \dot\xi_3 + h_3 h_1 \delta \dot\xi_1 + h_1 h_3 \delta \dot\xi_3 + h_2^2 \delta \dot\xi_1 + h_2^2 \delta \dot\xi_3 \\
& + 2 h_3 h_2 \delta \dot\xi_1 + h_3 h_2 \delta \dot\xi_3 + h_3^2 \delta \dot\xi_1 - h] B' A C = 0,
\end{aligned}
\tag{19}
$$

as well as

$$
\begin{aligned}
[&-h_1\delta\dot{\xi}_1 - h_2\delta\dot{\xi}_2 + h_3\delta\dot{\xi}_1 + h_3\delta\dot{\xi}_2 + (h_1^3 + 3h_1^2\delta\xi_1) + (h_2^3 + 3h_2^2\delta\xi_2) \\
&+ 2\,(h_3 h_1^2 + 2h_3 h_1\delta\xi_1 + h_1^2\delta\xi_3) + 2\,(h_3 h_2^2 + 2h_3 h_2\delta\xi_2 + h_2^2\delta\xi_3) \\
&+ 3\,(h_1^2 h_2 + 2h_1 h_2\delta\xi_1 + h_1^2\delta\xi_2) + 3\,(h_2^2 h_1 + 2h_2 h_1\delta\xi_2 + h_2^2\delta\xi_1) \\
&+ 3\,(h_3^2 h_1 + 2h_3 h_1\delta\xi_3 + h_3^2\delta\xi_1) + 3\,(h_3^2 h_2 + 2h_3 h_2\delta\xi_3 + h_3^2\delta\xi_3) \\
&+ 6\,(h_3 h_1 h_2 + h_1 h_2\delta\xi_3 + h_3 h_2\delta\xi_1 + h_3 h_1\delta\xi_2) - \delta\ddot{\xi}_1 - \delta\ddot{\xi}_2]ABC \\
&+ [h_1^2 + h_2^2 + h_1 h_3 + h_2 h_3 + 2h_1 h_2](\delta\dot{\xi}_1 A'BC + \delta\dot{\xi}_2 AB'C) \\
&- [h_1^2\delta\dot{\xi}_2 + h_1 h_2\delta\dot{\xi}_1 + h_1 h_2\delta\dot{\xi}_2 + h_3 h_1\delta\dot{\xi}_1 + 2h_1 h_3\delta\dot{\xi}_2 \\
&+ h_2^2\delta\dot{\xi}_1 + 2h_3 h_2\delta\dot{\xi}_1 + h_3 h_2\delta\dot{\xi}_2 + h_3^2\delta\dot{\xi}_1 + h_3^2\delta\dot{\xi}_2 - h]C'AB = 0\,,
\end{aligned}
\tag{20}
$$

where

$$
\begin{aligned}
h =\ & (h_1^3 h_2 + 3h_1^2 h_2\delta\xi_1 + h_1^3\delta\xi_2) - (h_3 h_1^3 + 3h_3 h_1^2\delta\xi_1 + h_1^3\delta\xi_3) \\
& - (h_2^2 h_1^2 + 2h_2 h_1^2\delta\xi_2 + 2h_2^2 h_1\delta\xi_1) - 3\,(h_3 h_1^2 h_2 + 2h_3 h_1 h_2\delta\xi_1 + h_3 h_1^2\delta\xi_2 + h_1^2 h_2\delta\xi_3) \\
& - (h_3^2 h_1^2 + 2h_3 h_1^2\delta\xi_3 + 2h_3^2 h_1\delta\xi_1) - (h_2^3 h_1 + 3h_2^2 h_1\delta\xi_2 + h_2^3\delta\xi_1) \\
& - 3\,(h_3 h_1 h_2^2 + 2h_3 h_1 h_2\delta\xi_2 + h_3 h_2^2\delta\xi_1 + h_1 h_2^2\delta\xi_3) \\
& - 3\,(h_3^2 h_1 h_2 + 2h_3 h_1 h_2\delta\xi_3 + h_3^2 h_2\delta\xi_1 + h_3^2 h_1\delta\xi_2) - (h_3^3 h_1 + 2h_3^2 h_1\delta\xi_3 + h_3^3\delta\xi_1) \\
& - (h_2^3 h_3 + 3h_2^2 h_3\delta\xi_2 + h_2^3\delta\xi_3) - (h_3^2 h_2^2 + 2h_3 h_2^2\delta\xi_3 + 2h_3^2 h_2\delta\xi_2) \\
& - (h_3^3 h_2 + 3h_3^3 h_2\delta\xi_3 + h_3^3\delta\xi_2)\,.
\end{aligned}
\tag{21}
$$

This set of equations can be reduced further by taking the derivatives of the $A, B$ and $C$ where the derivative is required. Then, collecting the perturbation terms together yields:

$$
\begin{aligned}
(&-\delta\ddot{\xi}_2 - \delta\ddot{\xi}_3 + \delta\dot{\xi}_2(-h_1^3 - h_2 h_1^2 - 2h_3 h_1^2 - h_2 h_3 h_1^2 + h_2^2 h_1 + h_1 + h_2 h_3 h_1 + h_2^3 + h_2 h_3^2 + 2h_2^2 h_3) \\
&+ \delta\dot{\xi}_3\left(-h_1^3 - 2h_2 h_1^2 - h_3 h_1^2 + h_1 + h_3^3 - h_2^2 + 2h_2 h_3^2 - 3h_2 + h_2^2 h_3 - h_3\right) \\
&+ \delta\xi_1(-3h_2 h_1^3 - 3h_3 h_1^3 - 2h_2^2 h_1^2 - 2h_3^2 h_1^2 \\
&- 6h_2 h_3 h_1^2 - h_2^3 h_1 - h_3^3 h_1 - 3h_2 h_3^2 h_1 + 6h_2 h_1 - 3h_2^2 h_3 h_1 + 6h_3 h_1 + 2h_2^2 + 2h_3^2 + 6h_2 h_3) \\
&+ \delta\xi_2(-h_1^4 - 2h_2 h_1^3 - 3h_3 h_1^3 - 3h_2^2 h_1^2 - 3h_3^2 h_1^2 - 6h_2 h_3 h_1^2 + 3h_1^2 - h_3^3 h_1 - 2h_2^2 h_3 h_1 + 4h_2 h_1 \\
&+ 6h_3 h_1 + 3h_2^2 + 6h_2 h_3) + \delta\xi_3(-h_1^4 - 2h_3^2 h_1^3 - 3h_2 h_1^3 - 2h_3 h_1^3 - 3h_2^2 h_1^2 - 6h_2 h_3 h_1^2 + 3h_1^2 \\
&- h_2^3 h_1 - 3h_2 h_3^2 h_1 + 6h_2 h_1 - 2h_2^2 h_3 h_1 + 4h_3 h_1 + 3h_2^2 + 3h_3^2))(abc)\,e^{(h_1 + h_2 + h_3)t} = 0
\end{aligned}
\tag{22}
$$

$$
\begin{aligned}
(&-\delta\ddot{\xi}_1 - \delta\ddot{\xi}_3 + \delta\dot{\xi}_1\left(h_1^3 + h_2 h_1^2 + 2h_3 h_1^2 - h_2^2 h_1 + h_3^2 h_1 - h_1 - h_2^3 - h_2 h_3^2 + h_2 - 2h_2^2 h_3\right) \\
&+ \delta\dot{\xi}_3\left(-h_2^3 - 2h_1 h_2^2 - h_3 h_2^2 - h_1^2 h_2 + h_3^2 h_2 + h_2 + h_3^3 + 2h_1 h_3^2 + h_1^2 h_3 - h_3\right) \\
&+ \delta\xi_1(-h_2^4 - 2h_1 h_2^3 - 3h_3 h_2^3 - 3h_1^2 h_2^2 - 3h_3^2 h_2^2 - 6h_1 h_3 h_2^2 + 3h_2^2 - h_3^3 h_2 - 2h_1 h_3^2 h_2 \\
&+ 4h_1 h_2 - 3h_1^2 h_3 h_2 + 6h_3 h_2 + 3h_1^2 + 3h_3^2 + 6h_1 h_3) \\
&+ \delta\xi_2(-h_2 h_1^3 - 2h_2^2 h_1^2 - 3h_2 h_3 h_1^2 + 2h_1^2 - 3h_2^3 h_1 - 3h_2 h_3^2 h_1 + 6h_2 h_1 - 6h_2^2 h_3 h_1 \\
&+ 6h_3 h_1 - h_2^3 h_3 - 2h_2^2 h_3^2 + 2h_3^2 - 3h_2^3 h_3 + 6h_2 h_3) \\
&+ \delta\xi_3(-h_2^4 - 3h_1 h_2^3 - 2h_3 h_2^3 - 3h_1^2 h_2^2 - 3h_3^2 h_2^2 - 6h_1 h_3 h_2^2 + 3h_2^2 - h_1^3 h_2 - 2h_1 h_3^2 h_2 \\
&+ 6h_1 h_2 - 2h_1^2 h_3 h_2 + 4h_3 h_2 + 3h_1^2 + 2h_3^2 + 6h_1 h_3))(abc)\,e^{(h_1 + h_2 + h_3)t} = 0
\end{aligned}
\tag{23}
$$

$$(-\delta\ddot{\xi}_1 - \delta\ddot{\xi}_2 + \delta\dot{\xi}_1\left(h_1^3 + 2h_2h_1^2 + h_3h_1^2 + h_2^2h_1 - h_3^2h_1 - h_1 - h_3^3 - 3hh_2^2 - 2h_2h_3 + h_3\right)$$
$$+\delta\dot{\xi}_2\left(h_2^3 + 2h_1h_2^2 + h_3h_2^2 - h_3^2h_2 - h_2 - h_3^3 - 2h_1h_3^2 + h_3\right) \tag{24}$$
$$+\delta\xi_1(-h_3^4 - 2h_1h_3^3 - 3h_2h_3^3 - 3h_1^2h_3^2 - 3h_2^2h_3^2 - 6h_1h_2h_3^2 + 3h_3^2 - h_2^3h_3 - 2h_1h_2^2h_3$$
$$+4h_1h_3 - 3h_1^2h_2h_3 + 6h_2h_3 + 3h_1^2 + 3h_2^2 + 6h_1h_2)$$
$$+\delta\xi_2(-h_3^4 - 3h_1h_3^3 - 2h_2h_3^3 - 3h_1^2h_3^2 - 3h_2^2h_3^2 - 6h_1h_2h_3^2 - h_1^3h_3 - 3h_1h_2^2h_3$$
$$+6h_1h_3 - 2h_1^2h_2h_3 + 4h_2h_3 + 3h_1^2 + 3h_2^2 + 6h_1h_2)$$
$$+\delta\xi_3(-h_3h_1^3 - 2h_3^2h_1^2 - 3h_2h_3h_1^2 + 2h_1^2 - 2h_3^3h_1 - 9h_3h_2^2h_1 - 6h_2h_3^2h_1 + 6h_2h_1 + 6h_3h_1$$
$$-3h_2h_3^3 + 2h_2^2 - 2h_2^2h_3^2 + 3h_3^2 - h_2^3h_3 + 6h_2h_3))(abc)\,e^{(h_1+h_2+h_3)t} = 0\,.$$

These equations can be reduced to a single equation:

$$(-h_2^4 - 3h_1h_2^3 - 4h_3h_2^3 - 5h_1^2h_2^2 - 6h_3^2h_2^2 - 11h_1h_3h_2^2 + 8h_2^2 - 3h_1^3h_2 - 4h_3^3h_2 - 11h_1h_3^2h_2$$
$$+16h_1h_2 - 12h_1^2h_3h_2 + 18h_3h_2 - h_3^4 - 3h_1h_3^3 + 6h_1^2 - 5h_1^2h_3^2 + 8h_3^2 - 3h_1^3h_3 + 16h_1h_3)\delta\xi_1$$
$$+(-h_1^4 - 3h_2h_1^3 - 4h_3h_1^3 - 5h_2^2h_1^2 - 6h_3^2h_1^2 - 11h_2h_3h_1^2 + 8h_1^2 - 3h_2^3h_1 - 4h_3^3h_1 - 11h_2h_3^2h_1$$
$$+16h_2h_1 - 12h_2^2h_3h_1 + 18h_3h_1 - h_3^4 - 3h_2h_3^3 + 6h_2^2 - 5h_2^2h_3^2 + 2h_3^2 - 3h_2^3h_3 + 16h_2h_3)\delta\xi_2$$
$$+(-h_1^4 - 2h_3^2h_1^2 - 4h_2h_1^3 - 3h_3h_1^3 - 6h_2^2h_1^2 - 2h_3^2h_1^2 - 11h_2h_3h_1^2 + 8h_1^2 - 4h_2^3h_1 - 2h_3^3h_1$$
$$-11h_2h_3^2h_1 + 18h_2h_1 - 11h_2^2h_3h_1 + 16h_3h_1 - h_2^4 - 3h_2h_3^3 + 8h_2^2 - 5h_2^2h_3^2 + 8h_3^2$$
$$-3h_2^3h_3 + 10h_2h_3)\delta\xi_3 + \alpha_1\delta\dot{\xi}_1 + \gamma_1\delta\dot{\xi}_2 + \kappa_1\delta\dot{\xi}_3 - 2(\delta\ddot{\xi}_1 + \delta\ddot{\xi}_2 + \delta\ddot{\xi}_3) = 0 \tag{25}$$

where

$$\alpha_1 = 2h_1^3 + 3h_2h_1^2 + 3h_3h_1^2 - 2h_1 - h_2^3 - h_3^3 - h_2h_3^2 + h_2 - 3h_2^2h_3 - 2h_2h_3 + h_3 \tag{26}$$

$$\gamma_1 = -h_1^3 - h_2h_1^2 - h_2h_3h_1^2 - 2h_3h_1^2 + 3h_2^2h_1 \tag{27}$$
$$-2h_3^2h_1 + h_2h_3h_1 + h_1 + 2h_2^3 - h_3^3 - h_2 + 3h_2^2h_3 + h_3$$

$$\kappa_1 = -h_1^3 - 3h_2h_1^2 - 2h_2^2h_1 + 2h_3^2h_1 + h_1 - h_2^3 + 2h_3^3 - h_2^2 + 3h_2h_3^2 - 2h_2 - 2h_3\,. \tag{28}$$

The above equations 22-24 are long ordinary differential equations in terms of $\delta\xi$'s and their first and second derivatives. After solving them we get:

$$\delta\xi_1 = c_1\exp\left(\frac{1}{2}\alpha t\right) + c_2\exp\left(\frac{1}{2}\beta t\right) \tag{29}$$

$$\alpha = \frac{\alpha_1}{4} - \sqrt{\alpha_2 + \left(\frac{\alpha_1}{4}\right)^2} \tag{30}$$

$$\beta = \frac{\alpha_1}{4} + \sqrt{\alpha_2 + \left(\frac{\alpha_1}{4}\right)^2} \tag{31}$$

$$\alpha_2 = -h_2^4 - h_1h_2^3 - 4h_3h_2^3 - h_1^2h_2^2 - 6h_3^2h_2^2 - 5h_1h_3h_2^2 + 4h_2^2 \tag{32}$$
$$+3h_1^3h_2 - 4h_3^3h_2 - 5h_1h_3^2h_2 + 4h_1h_2 + 6h_3h_2 - h_3^4 - h_1h_3^3 + 6h_1^2 - h_1^2h_3^2 + 4h_3^2$$

$$\delta\xi_2 = c_3\exp\left(\frac{1}{2}\gamma t\right) + c_4\exp\left(\frac{1}{2}\zeta t\right) \tag{33}$$

$$\gamma = \frac{C_1}{4} - \sqrt{\gamma_2 + \left(\frac{\gamma_1}{4}\right)^2} \tag{34}$$

$$\zeta = \frac{C_1}{4} + \sqrt{\gamma_2 + (\frac{\gamma_1}{4})^2} \tag{35}$$

$$\begin{aligned} \gamma_2 &= -h_1^4 - h_2 h_1^3 - 4h_3 h_1^3 - h_2^2 h_1^2 - 6h_3^2 h_1^2 - 5h_2 h_3 h_1^2 + 4h_1^2 \\ &\quad + 3h_2^3 h_1 - 4h_3^3 h_1 - 5h_2 h_3^2 h_1 + 4h_2 h_1 + 6h_3 h_1 - h_3^4 - h_2 h_3^3 + 6h_2^2 - h_2^2 h_3^2 - 2h_3^2 \end{aligned} \tag{36}$$

$$\delta\xi_3 = c_5 \exp\left(\frac{\kappa_2}{\kappa_1} t\right) \tag{37}$$

$$\begin{aligned} \kappa_2 &= -h_1^4 - 2h_3^2 h_1^3 - 4h_2 h_1^3 - h_3 h_1^3 - 6h_2^2 h_1^2 + 2h_3^2 h_1^2 - 5h_2 h_3 h_1^2 + 4h_1^2 \\ &\quad -4h_2^3 h_1 + 2h_3^3 h_1 + h_2 h_3^2 h_1 + 6h_2 h_1 - 5h_2^2 h_3 h_1 + 4h_3 h_1 \\ &\quad -h_2^4 + 3h_2 h_3^3 + 4h_2^2 - h_2^2 h_3^2 + 2h_3^2 - h_2^3 h_3 - 2h_2 h_3 \; . \end{aligned} \tag{38}$$

In the above equations, provided that $\alpha, \gamma < 0$ is satisfied, if $\frac{\alpha_1}{4}, \frac{\gamma_1}{4} < 0$, $|\frac{\alpha_1}{4}| > |\alpha_2|$, $|\frac{\gamma_1}{4}| > |\gamma_2|$ and either $\kappa_1 < 0$ or $\kappa_2 < 0$, but not both, then if we substitute very large numbers for $t$; (i.e $t \to \infty$) we find out that $\delta\xi_i \to 0$ since the powers of the exponents become negative.

**Example**

The $h's$ can be considered as ratios of original Hubble constant with respect to the current one. Substituting random values of $h's < 1$, e.g. $h_1 = 0.9, h_2 = 0.5, h_3 = 0.2$, will lead to the following solutions:

$$\begin{aligned} \delta\xi_1 &= e^{-1.56302t}\left(c_1 + c_2 e^{3.64472t}\right) \\ \delta\xi_2 &= e^{-1.36781t}\left(c_3 + c_4 e^{2.78428t}\right) \\ \delta\xi_3 &= e^{-3.87861t}\left(c_5 + c_6 e^{0.757611t}\right) \; . \end{aligned} \tag{39}$$

The above perturbation terms converge to zero as $t \to \infty$ provided that $c_2, c_4 = 0$. If we make other substitutions then $c_6 = 0$ must be also satisfied. Substituting other values can lead to similar results if the second terms in the perturbation equations are ruled out. Hence, we can generalize that if $c_{2,4,6} = 0$ the perturbation terms can diminish. Consequently, we can deduce that the de Sitter universe is stable against the perturbations. Therefore, it is stable in the context of the Nash gravity. Otherwise, the de Sitter universe will become unstable as the perturbations become very large in the context of Nash gravity.

# 7 Conclusion and future work

As an alternative theory for Einsteins theory of gravity another theory was developed and modified by John Nash. The theory is divergence free and considered to be of interest in constructing theories of quantum gravity. In this paper, we examine the Einstein static and the de Sitter Universe solutions and quantify their stabilities for the Bianchi type I model using Nash modified theory of gravity in the context of Bianchi Universe geometry. We searched for the stability of Einstein and de Sitter Universes using Nash gravity Model. We used the metric of Bianchi type I model in the Nash field equation. Then we substituted the appropriate scale factors in the field equations depending on the type of Universe under study. We have seen that the Einstein static Universe leads to instabilities against perturbation in the Hubble

constant. This implies the inadequacy of the Einstein static Universe in its current format mingled with the Nash theory of gravity to describe the eternal Universe against instabilities unless some modification is implemented in either theories. Further studies can be achieved that can lead to a potential stability for the eternal Universe. On the other hand the de Sitter Universe shows stability against the perturbation with a certain parametric combination or setting some constant terms which result from the differential equation to zero. This stability can be investigated in further studies with other Bianchi types, for example. If it works in other models it can be a very good framework to explain the Universe with no divergence. Also, it can help us to understand the inflation era and the subsequent periods. On the other hand, the theoretical limits obtained for the values of the directional Hubble parameters help to understand the anisotropy in the Universe.

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
