# Peer review of "Linear Stability of Einstein and de Sitter Universes in the Quadratic Theory of Modified Gravity"

_SciPost Physics, doi:SciPost Phys. Proc. 4, 012 (2021)_

## Round 1 · Referee Report · Anonymous · 2020-11-23

Strengths

1) The problem under study and the method of solution are clearly stated.

Weaknesses

1) There are frequent typos and grammatical errors
2) Some of the equations are related by symmetry so instead of repeating them three times they could have been better organised to only appear once.
3) The results are just summarised in the conclusions without discussing where they leave us in terms of the relevance of the given theory for cosmology and no outlook for future studies is provided.
4) Although the reference list quite extensive, some standard references, for instance to the early work by Stelle on similar theories, could perhaps also have been included.

Report

The paper is appropriate for publication in SciPost Physics proceedings as it reports on a contribution to the relevant conference.

The work examines the appropriateness for cosmology of an alternative theory of gravity which is purely quadratic in the curvature tensors. In particular, it studies the linear stability of the Einstein static universe and de Sitter space in this theory. As in standard GR, but unlike other modifications of gravity which include both the standard and higher derivative terms, the Einstein static universe appears to be unstable. Assuming the validity of certain conditions stated in the paper, de Sitter space appears to be stable under linear perturbations.

This article is part of an ongoing effort to understand the validity and relevance of various modified gravity theories to current questions in cosmology, and as such can help to guide future studies.

The work can be published after the author addresses the changes/comments below.

Requested changes

1) In several places, "anistropic" should be replaced by "anisotropic"
2) In section 3: non-isotroposity->anisotropy (or non-isotropy?)
3) Above eq.16: "yield to"->"arrive at"
4) above eq.28: "redeuced"->"reduced"
5)The derivation and meaning of (28) is not clear. In particular, should the $\xi$'s be $\delta\xi$'s? Also the relevance of this equation is not clear as it doesn't appear to be necessary for the discussion that follows.
6) In (43) there is probably a typo as the first term on the third line contains an $h$ which should be either $h_1$,$h_2$ or $h_3$.

7) End of section 6: The main result of the contribution hinges on the parameters $A_{1,2},C_{1,2},E_{1,2}$ being positive at the same time. Since these are complicated functions of the parameters $h_i$, it is not immediately clear that such a solution exists. A proof of this claim, or an example of a domain where it is true, should be included here.

8) A related confusing point is that the lagrangian (5) and the following equations of motion (6)-(8) are symmetric under exchange of the 1,2,3 labels, and the fluctuation parameters are also introduced in a symmetric way at the beginning of section 6. So one would expect the final answer for the fluctuations (29),(34),(39) to reflect that symmetry, in other words the $A_{1,2},C_{1,2},E_{1,2}$ parameters to be mapped to each other under suitable exchanges of the $h_i$ parameters. This does not seem to be the case. Is it possible that some terms are missing in (29),(34),(39)? Otherwise, the author should comment on where this symmetry breaking arose during the process of solving the fluctuation equations.

9) In the conclusions, the relevance of the stability condition mentioned in point 7) above should be discussed further. Is it a natural condition and what are its implications for the validity of this alternative gravity theory? Similarly, do we learn anything from the instability for the Einstein static universe, since this is not a valid cosmological solution anyway? A short outlook could also be included.

  • validity: ok
  • significance: ok
  • originality: good
  • clarity: high
  • formatting: reasonable
  • grammar: acceptable

Author:  Mudhahir Al-Ajmi  on 2021-03-04

(in reply to Report 1 on 2020-11-23)
Category:
answer to question
correction
validation or rederivation
suggestion for further work

Dear Editor, Thank you very much for your valuable comments and remarks. I tried hard to make the paper up to the level of your satisfaction. List of the changes are explained below: 1. The typos mentioned are corrected plus others, 2. The change has been made from "non-isotroposity" to "anisotropy" as suggested. 3. The term "arrive at" is used instead of "yield to" as suggested thankfully. 4. The word "redeuced" to "reduced". 5. Equation mentioned (eq 28) has been removed to clear our to confusion that it may cause. 6. Equation 43 is corrected. We have done a thorough revision of equations as follows: a. Equations in the previous versions were too long and somewhat boring. The equations (starting from Equation 6) are written in a shorter format and the repeated terms are written separately (equation 9, 17, 21, 27-29 in the updated version). b. In Section 6 in equation 27 (in the previous version, equation 24 in the updated version), there was a typo of h as mentioned by the referees. This also has been corrected. This leads to formulating the subsequent equations again accordingly (equations 25-38). 7. The parameters A_1, A_2, C_1, C_2, E_1, E_2 are rewritten again (as explained in point 6b above). The constant A,B,C A_1, A_2, C_1, C_2, E_1, E_2 are replaced by Greek letters instead to avoid the confusion with the earlier notations used for the scale factors. An example of our claim is given in a subsection. The above mentioned parameters are rewritten again because of the reason mentioned in 6b. 8. I rechecked equation 5-8. They seem to be OK. We think that, after substituting the perturbation terms and because we are omitting the higher order perturbation terms, somewhat the mentioned symmetry in the equation has been lost specially we can see that the Lagrangian (eq 5) is not highly symmetric. 9. The conclusion is modified and few sentences are added as future works. The updated version of the file is attached. Thank you, again, for your fruitful comments which led, undoubtfully, to the improvement of the search. Mudhahir Al-Ajmi

Attachment:

Stability_Project.pdf

---

## Round 2 · Referee Report · Anonymous (Referee 2) · 2021-4-14

Report

The author has addressed the main concerns raised in the first submission and in particular provided a more detailed discussion of the requirements for the fluctuation exponents to be negative, including an example as requested. The conclusions were slightly modified accordingly to indicate that stability in the de Sitter case depends on certain choices of constants (which perhaps can be further investigated in the future).

In addition, several typos were corrected as requested and many clarifications were made.

Therefore the submission can now be published in SciPost Physics Proceedings.

---

## Round 2 · Author Response

Dear Editor, Thank you very much for your valuable comments and remarks. I tried hard to make the paper up to the level of your satisfaction. List of the changes are explained below: 1. The typos mentioned are corrected plus others, 2. The change has been made from "non-isotroposity" to "anisotropy" as suggested. 3. The term "arrive at" is used instead of "yield to" as suggested thankfully. 4. The word "redeuced" to "reduced". 5. Equation mentioned (eq 28) has been removed to clear our to confusion that it may cause. 6. Equation 43 is corrected. We have done a thorough revision of equations as follows: a. Equations in the previous versions were too long and somewhat boring. The equations (starting from Equation 6) are written in a shorter format and the repeated terms are written separately (equation 9, 17, 21, 27-29 in the updated version). b. In Section 6 in equation 27 (in the previous version, equation 24 in the updated version), there was a typo of h as mentioned by the referees. This also has been corrected. This leads to formulating the subsequent equations again accordingly (equations 25-38). 7. The parameters A_1, A_2, C_1, C_2, E_1, E_2 are rewritten again (as explained in point 6b above). The constant A,B,C A_1, A_2, C_1, C_2, E_1, E_2 are replaced by Greek letters instead to avoid the confusion with the earlier notations used for the scale factors. An example of our claim is given in a subsection. The above mentioned parameters are rewritten again because of the reason mentioned in 6b. 8. I rechecked equation 5-8. They seem to be OK. We think that, after substituting the perturbation terms and because we are omitting the higher order perturbation terms, somewhat the mentioned symmetry in the equation has been lost specially we can see that the Lagrangian (eq 5) is not highly symmetric. 9. The conclusion is modified and few sentences are added as future works. The updated version of the file is attached. Thank you, again, for your fruitful comments which led, undoubtfully, to the improvement of the search. Mudhahir Al-Ajmi

---

## Round 2 · List of Changes

1. The typos mentioned are corrected plus others,
2. The change has been made from "non-isotroposity" to "anisotropy" as suggested.
3. The term "arrive at" is used instead of "yield to" as suggested thankfully.
4. The word "redeuced" to "reduced".
5. Equation mentioned (eq 28) has been removed to clear our to confusion that it may cause.
6. Equation 43 is corrected. We have done a thorough revision of equations as follows: a. Equations in the previous versions were too long and somewhat boring. The equations (starting from Equation 6) are written in a shorter format and the repeated terms are written separately (equation 9, 17, 21, 27-29 in the updated version). b. In Section 6 in equation 27 (in the previous version, equation 24 in the updated version), there was a typo of h as mentioned by the referees. This also has been corrected. This leads to formulating the subsequent equations again accordingly (equations 25-38).
7. The parameters A_1, A_2, C_1, C_2, E_1, E_2 are rewritten again (as explained in point 6b above). The constant A,B,C A_1, A_2, C_1, C_2, E_1, E_2 are replaced by Greek letters instead to avoid the confusion with the earlier notations used for the scale factors. An example of our claim is given in a subsection. The above mentioned parameters are rewritten again because of the reason mentioned in 6b.
8. I rechecked equation 5-8. They seem to be OK. We think that, after substituting the perturbation terms and because we are omitting the higher order perturbation terms, somewhat the mentioned symmetry in the equation has been lost specially we can see that the Lagrangian (eq 5) is not highly symmetric.
9. The conclusion is modified and few sentences are added as future works. The updated version of the file is attached. Thank you, again, for your fruitful comments which led, undoubtfully, to the improvement of the search.

---

## Editorial Decision

published